# Coming of Age: Cryo-Electron Tomography as a Versatile Tool to Generate High-Resolution Structures at Cellular/Biological Interfaces

**DOI:** 10.3390/ijms22126177

**Published:** 2021-06-08

**Authors:** Zuoneng Wang, Qingyang Zhang, Carsten Mim

**Affiliations:** Department of Biomedical Engineering and Health Systems, Royal Technical Institute (KTH), Hälsovägen 11C, 141 27 Huddinge, Sweden; zuonewan@kth.se (Z.W.); qinzha@kth.se (Q.Z.)

**Keywords:** protein complexes, structural biology, electron microscopy

## Abstract

Over the last few years, cryo electron microscopy has become the most important method in structural biology. While 80% of deposited maps are from single particle analysis, electron tomography has grown to become the second most important method. In particular sub-tomogram averaging has matured as a method, delivering structures between 2 and 5 Å from complexes in cells as well as in vitro complexes. While this resolution range is not standard, novel developments point toward a promising future. Here, we provide a guide for the workflow from sample to structure to gain insight into this emerging field.

## 1. Introduction

### 1.1. Structural Biology of Macro-Molecular Complexes

Biological structures, from organs and tissues to cells and molecules, are major determinants of biological function. This deterministic/teleonomic view is the basic tenet of structural biology [1]. Today we have an arsenal of techniques to generate structures. Medical (X-ray and MRI) scanners capture organs and tissues. Light microscopic techniques image tissues and cells. Finally, structure biological methods determine the structures of proteins and the complexes they form. 

The moment protein structures became available, the idea that proteins are molecular machines, with sidechains as cogwheels, became tangible. One of the first examples was the visualization of cooperativity in hemoglobin, where the movement of residues in another subunit influences the activity of the catalytic center [2,3]. Structures do not exist in a vacuum but are part of a functional network. Yet, structures, even if they are related, may confer different biological activities [4]. Therefore, the context of a structure within its functional unit is important to understand its role and *vice versa*.

Traditional structural biological techniques are examples of a reductionist approach mostly to reduce the complexity of the investigated system. Single molecules, even if they exist in heteromeric assemblies, are isolated from expression systems and rarely from natural sources. X-ray crystallography requires high concentrations and high purity of the investigated sample. Because crystallization is sensitive to environmental factors, progress was restricted for a long time. Membrane proteins were particularly challenging, as well as large heterogenous complexes; this changed when the structure of the photosynthetic reaction center became available in 1985 [5]. Structures of megadalton-size molecular machines, like the ribosome, were solved even later [6]. Although impressive, these structures are still highly pure samples in highly optimized buffers. Visualizing ligand interactions (in the same crystal) is possible in X-ray crystallography, as long as they do not impair the order/integrity of the crystal. Otherwise, the crystallization process must be optimized again.

Nuclear Magnetic Resonance spectroscopy (NMR) offers benefits over X-ray crystallography. NMR can deliver a dynamic picture of interactions within the sample. This opens the possibility of studying the interaction of proteins with other biomolecules, like lipids, nucleotides or other ligands in situ. The study of intrinsically disordered parts of a protein is possible as well. Structural determination of a sample is possible in solution [7] as well as in solid state [8]. The latter presents the opportunity to study long polymers like fibrils. However, NMR requires high concentrations of the sample and, in the case of solution state NMR, high solubility of the sample. The feasibility of resonance peak assignment restricts the size of the investigated protomer. Specific labelling of amino acids, residues or atoms allows the investigation of larger complexes, but *de novo* structures of (>100 kD) large (single) proteins is still out of reach. However, NMR has enabled us to obtain a dynamic picture of proteins with their (near) natural interfaces, like membranes [9].

In order to understand the function of complexes, we have to image them in the context of their natural environment. Light microscopy has made big strides in recent years with methods that go beyond the diffraction limit of the light. These techniques produce stunning images of cells and organelles and their sub-structures at resolutions between 10 and 70 nm; however, sub 10 nm resolution is possible [10]. There is a resolution gap between light microscopy and X-ray crystallography/NMR. Fortunately, this resolution (and visualization) gap can be bridged by electron microscopy (Figure 1A).

Electron microscopy (EM) is an imaging technique that encompasses a variety of applications, some of which are elaborated herein. The main advantage of EM is the fact that electron microscopes produce images from which high-resolution structures are extracted. Of special interest is transmission electron microscopy, where images are 2D projections of a 3D sample [11]. Because EM is generating images, a wide range of samples can be investigated by EM, from single (protein) molecules to organelles and cells (Figure 1).

EM has seen immense growth. Since 2008, the number of maps has doubled every two years (e.g., see Figure 1). As a result, this technique has received wide recognition in the public and industry. Of the entries into the EM database (emdb.org), 80% are maps from single particle analysis (SPA) studies. Advances in data acquisition, image processing and analysis resulted in structures that routinely achieve a resolution of 3–5 Å; however, <2 Å structures are possible [12]. The imaged samples can be as small as 64–52 kDa [13,14]. It is difficult to keep track of which size or resolution record has been broken, but it is safe to say that <100 kDa proteins can be resolved with affordable instruments to resolutions that allow *de novo* model building [15]. For a detailed review of this technique, see other articles in this issue. While the above achievements deserve praise, the full potential of electron microscopy as a technique remains veiled. EM can image multiple states of a protein simultaneously, which can be resolved and placed in an energetic landscape [16,17]. More importantly, sample preparation allows a broad range of specimens to be imaged, allowing for the structural determination of proteins from natural sources. This may lead to the revision of findings from structures isolated out of sophisticated expression systems [18]. For SPA-related studies, it is important that the particles exhibit relative uniformity or order. This requirement is particularly true for the investigation of macromolecular assemblies like protein-decorated filaments or viral envelopes.

Although SPA alleviates some of the shortcomings of other structural biology methods, namely the relative ease with which molecular machines can be imaged, SPA samples are still derived from complex purification schemes. Each purification scheme selects for the most stable complexes. There are means to stabilize more transient complexes for EM [19]. Purification also removes the investigated sample from its natural interaction and functional network. As a result, transient interactions or weak interactions, e.g., to lipids and cofactors, get lost. The protein interaction network contains not only information about binding partners but also information about the localization within the cell or compartment. Unless specific markers can be retrieved, cellular localization information is permanently lost during purification.

### 1.2. Electron Tomography 

The reconstruction of a 3D structure out of 2D (projection) images hinges on the determination of the angles between projections from particles (or their averages). While crystallographic order was used to solve the first structure of a multi-protein complex by EM, the authors have already suggested that the collection of the same particle at different tilt angles can lead to a structure and can be expanded to “unsymmetrical particles, or sections of biological specimens” [11]. Collecting images at predetermined angles is one of the earliest methods of generating an SPA *de novo* structure without prior knowledge of the sample [20]. The real power of this approach is the fact that it can be applied to any sample, and it will result in a 3D structure, even if there is only a single copy of the specimen in the image. Yet, if the specimen has low or no symmetry, a minimum number of (tilted) views is necessary, and the achievable resolution depends on the total number of tilted images [11,21]. If we want to generate images at different tilt angles (tilt series) and calculate a structure from the tilt series (electron tomogram), we need to overcome multiple challenges (Figure 2). First, the specimen has to stay at the center of the image during data collection so one can align the same structure over all tilt angles. Second, the sample has to be thin enough to be penetrated by the beam, even at high tilt angles. Third, the imaged sample has to be beam resistant, because it has to endure multiple exposures. The latter is an inherent problem of biological samples. One solution to this problem is embedding the sample in heavy metal salts. However, staining is not a perfect solution; the grain size of the stain limits the ultimate resolution of the tomogram, the sample still experiences damage [22,23], and areas that are inaccessible to the stain are “invisible”. The discovery of sample vitrification and the preservation of biological samples at cryogenic temperatures [24] made it possible to collect tilt series to create cryo-electron tomograms (CT). The next advance was the introduction of charge coupled device (CCD) cameras, which allowed the automatization of the different steps in the collection of a tilt series (tilting, tracking) [25]. However, the accumulation of electrons in the sample over the collection of a whole tilt series (30–40 images) is still a problem. Doses of 120–160e^−^/Å^2^ cause extensive damage (visible by bubbling of the ice [23]) and the loss of high-resolution features, e.g., decarboxylation of side chains [26]. A proof of concept study showed that it is possible to image at low doses and reconstruct samples in vitreous ice [27]. It took another five years until the first cryo-tomograms of organelles and organelle-derived vesicles were reported [28,29]. With this technique at hand, electron microscopy is able to investigate macromolecular machines in reconstituted systems (in vitro), isolated from cells (ex situ), or inside cells (in situ). The same advances that propelled SPA to where it stands today benefitted cryo-electron tomography (cryoET). This is reflected by the increase of cryo-tomograms submitted (Figure 1C) as well as the drop of the resolution from cryoET derived structures (Figure 1B). This is true for in situ cryoET as well as cryoET for other samples (Figure 3). The ease with which CTs are recorded and reconstructed makes cryoET a quality-control tool to assess the accumulation of proteins at the air-water interface and the ice thickness [30]. 

In the following sections, we present an overview workflow in cryoET, from sample preparation after data acquisition to data processing and interpretation.

## 2. Consideration for the Collection of Cryo-Electron Tomograms

### 2.1. Sample Preparation in CryoET

Generally, samples in cryoET are either reconstituted specimens from purified components (e.g., protein bound to microtubules or membranes) or cells and cell components (Figure 3). In either case, the thickness of the sample is a concern, because the mean free path for inelastic scattering is about 300–400 nm (for a 300 kV microscope) [31]. Logically, at higher tilt angles the ice thickness increases (along the beam axis). Thicker ice increases the likelihood of inelastic scattering, which increases background noise. Therefore, thinner samples yield better data (ideally <100 nm). Yet, the ice layer should encompass the structure of interest. The latter is obviously a concern for cells or cell-derived samples, as many substructures exceed 100 nm thickness.

Samples up to 5 µm thickness can be vitrified [24] with plunge freeze devices and thinned later if necessary (Figure 4). With these requirements, dispersed molecular complexes fare better, because blotting parameters can be adjusted to create ice as thin as possible. In vitro samples like this have achieved sub 5 Å resolution, either as part of assemblies [32,33] or as single copies [34,35] (see Figure 3). The latter is as close to the resolution of the same (~300 kDa) particle as solved with SPA.

The ultimate goal of cryoET is the structural study of proteins in a natural/physiological environment. To avoid the confining thickness of cells, model systems are used to maintain part of the native interaction network for structural studies. Structural studies can be conducted with bacteria [36], bacterial mini cells [37,38], protrusion of cells (e.g., [39]), fractionated cellular compartments like mitochondria [40], flagella [41], various vesicles [42,43] and viruses [44,45]. This list is not complete but shows the options researchers have without being limited by complete (eukaryotic) cells.

If we want to image isolated tissues, cells or cell compartments, we have to overcome two potential issues. First, if the sample is thicker than 5 µm, plunge freezing does not ensure the generation of vitreous ice throughout the sample. High-pressure freezing is the method of choice for larger specimens (e.g., [46]). This method is able to cryo-preserve tissue samples or even small organisms like *C. elegans* [47]. Second, regardless of the freezing method, a thinning of the sample is necessary to make it suitable for cryoEM. CryoEM Of Vitreous Sections (CEMOVIS) was the first solution to generate samples thin enough for cryoET [48]. In short, the sample is cooled to maintain water in vitreous form and then cut with a diamond blade to obtain a thin film. The film is deposited on a grid and imaged. This method has two disadvantages: the cutting procedure generates surface artefacts and the blade compresses the sample. Recently, a modified method, using re-vitrification of sections, delivered in situ structures of viruses in cells with a resolution of ~5 nm [49]. Another sample preparation method takes advantage of an approach used in material science. Here, a focused ion beam (FIB) of sputtered heavy ions is used to remove layers of the sample [50]. Dual beam instruments are used to monitor the process (and image the sample). The method has been improved to generate lamellae in different shapes and positions within cells (Figure 4, box). The value of this approach was demonstrated by imaging and reconstructing COPI vesicles and the cellular machinery around amyloid fibers in cells, just to name two examples [51,52]. In both cases it was possible to localize, identify and classify COPI coats and proteasomes. At a pixel size of about 3.5 Å, the imaged area is about 1 µm × 1 µm. Thus, it is crucial to identify areas of interest for FIB milling or imaging. Fluorescence microscopy under cryo conditions is now routinely used to target areas for imaging on the grid [53,54]. Due to sample restrictions, standard correlative light and electron microscopy (CLEM) setups are no more than tools to find areas of interest. However, efforts to take advantage of high-resolution light microscopic methods like cryoSIM or Cryo-SOFI can help to interpret cryoET data in the larger context of the cell [55,56].

### 2.2. Data Collection

Automatizing the collection of tilt series has been an important step in the development of cryoET. This achievement would have not been possible without the introduction of coupled charge device (CCD) cameras during the time when photographic film was the medium of choice. CCD cameras digitize the signal immediately, but they have three significant shortcomings compared to photographic film: first, the detective quantum efficiency (DQE) is lower than photographic film; second, the recorded area is fairly small compared to film; third, CCD cameras do not detect electrons directly. Due to the lower dose per image and the scarcity of unique particles per image, cryoET was more limited by CCD cameras than SPA. But cryoEM was revolutionized by the introduction of complementary metal oxide semiconductor (CMOS) cameras with direct electron detectors (DEDs). The benefit of these cameras is twofold: a fast readout and a higher DQE (for a comparison of different recording media, see [57]). The high frame rate allows the fractionation of electron dose over many frames and the correction of sample movement (Figure 2B). Multiple frame collection (i.e., movies) has been crucial in SPA to solve maps of proteins to atomic resolution. The case of cryoET dose fractionation during the collection of tilt series is more challenging. The overall dose per image in a tilt series is about 10–20 times lower compared to the single exposure of an SPA image. Thus, the initial alignment of single frames is more difficult and relies on fiducial markers (like colloidal gold); therefore, a high frame rate with current instrumentation is not beneficial.

Because the resolution of cryoET depends on the number of views (or images at different tilt angles), it is desirable to take as many images as possible. However, mechanical limitations of the specimen stage and the total permissible radiation dose restrict experimental options. Most microscopes are able to tilt ±60°. Due to these restrictions in the tilt geometry, tilt series miss data (missing wedge). This could be mitigated by tilting the sample in X and Y. However, this solution is not feasible for cryoET due to the accumulated dose. Traditionally, tilt series were collected in ascending (or descending) increments (e.g., from −60° to 60° or *vice versa*) to minimize the stage displacement from one tilt angle to the next, which reduces the time needed to track stage errors. However, the image quality suffers from low to high tilt angles due to the increase in ice thickness. As a consequence, the sample is exposed to electrons without generating images with high-resolution data. Therefore, Hagen et al. devised a scheme to collect data at low tilt angles first, before collecting high tilt angle data [58]. The trade-off is a slower pace of data collection and the necessity of a high precision stage (e.g., to reduce the tracking error). While camera speed has increased dramatically, most of the time during a tilt series is spent moving and tracking the stage. Modern stages, dedicated to cryoET, are more precise, which allows data collection schemes that skip stage corrections. As a result, tilt series can be collected within 5 min [59,60]. It is not clear if these cryoETs are suitable for high-resolution structure generation. Bouvette et al. introduced an interesting scheme. The stage corrections are still performed for each tilt, but multiple positions are imaged with beam-induced image-shift [34]. The resulting cryoETs are of high quality and allow structures of smaller (300 kD) proteins to be solved to a resolution <4 Å.

DED cameras and motion correction improve the signal-to-noise ratio (SNR) after the image has been formed in the microscope. The generation of contrast in a microscope is mathematically described by the contrast transfer function (CTF). Usually, contrast is generated by defocusing the microscope. Due to the low SNR in tilt series, the defocus values are high (>3–6 µm). As a consequence of the CTF, the signal is subjected to a low-pass filter (for the modelling of the CTF see, e.g., [61]). To make things worse, cryoET samples are generally thicker than SPA samples, and thus inelastically scattered electrons are more likely. Removing them with appropriate devices (energy filters) improves the contrast of images significantly [62]. Energy filters are either part of the microscope (in-column filters) or after the microscope and in front of the camera. Another technical device that enhances the contrast is the phase plate (for a review, see [63]). Currently, the most common phase plate is the volta phase plate (VPP). VPPs have been used for the collection of SPA data as well as cryoET. However, the verdict on the benefit of VPPs for high-resolution structures, for instance for G-protein coupled receptors, is still out (e.g., [64]). The VPP also needs a precise adjustment of the defocus, which can be a problem for the collection of a tilt series due to the defocus gradient along the tilted sample.

Automated data collection is implemented in commercial data collection suites from EM and camera manufacturers (ThermoFisher, GATAN, TVIPS). Alternatively, there is an abundance of free software packages like SerialEM [65], TOM [66], UCSF Tomography [67] and Leginon [68]. This list is not exhaustive and does not include scripts or modifications to the software to implement features like the ones above (e.g., [34,59]).

## 3. Data Processing

### 3.1. Tomogram Generation

After pre-processing each individual image of a tilt series needs to be precisely aligned. Colloidal gold particles (fiducial markers) are usually added to the sample before freezing. In specimens that contain no fiducial markers, like FIB milled lamella, patch-tracking can be used for the alignment [69,70]. The reduced alignment precision in patch-tracking leads to a lower resolution of the tomograms, but there are methods to add fiducial markers to the cells (e.g., [71]) or lamellas after FIB milling [72]. The alignment is a crucial step and is done iteratively. If segmented volumes of the tomogram (sub-tomograms) are processed, it is possible to use these volumes as fiducial markers to correct for local deformations [35,73]. The phase contrast generated by the microscope introduces artefacts in images. Therefore tomograms must be corrected after estimating the CTF by programs like CTFFIND4 [61] or CTFPLOTTER [74]. The CTF is not uniform over the whole image at any given tilt, because the defocus value changes away from the tilt axis (with the exception of the 0° image). The image can be CTF corrected in 2D by dividing it into tiles or stripes and calculating the defocus gradient [74,75,76]. For CTF correction of high-resolution structures, the position of the sub-tomogram within the tomogram must be considered [77]. Individual sub-tomogram CTF correction is possible as well [35,73]. Once the tilt series is aligned and all other parameters are determined (shifts, precise tilt angles, offsets) the tomogram can be reconstructed from the tilt series. The most common approach is weighted back-projection, which preserves high-resolution data [20,78]. Although other more recent methods exist, some of which compensate for the missing wedge, their usefulness for high-resolution work needs to be tested [79,80,81].

### 3.2. Sub-Tomogram Averaging 

The greatest strength of cryoET is the instant generation of 3D structures. If repetitive structures or multiple copies of a complex are present, sub-volumes (or sub-tomograms) of the particles can be averaged and classified (see Figure 5 for a general workflow). Sub-tomogram averaging (STA) is a popular technique, accounting for 8% of the maps in the EMDB. This is more than the maps solved by helical reconstruction and electron crystallography combined. A sub-tomogram averaging project starts with the identification and extraction of sub-tomograms (STs). The STs are found by template matching of a known structure or by generating a template from manually picked sub-tomograms [52,82]. Alternatively, a grid can be constructed, and sub-tomograms are extracted in defined increments. The latter is used on filaments and assemblies on membranes (e.g., [32,33]). Machine learning algorithms like tensor voting can help with automatic segmentation [83]. Convoluted neuronal networks can also be trained to detect sub-tomograms without prior structural information [84]. The STs are then aligned with respect to one or multiple references. The alignment of the ST is performed on the basis of a similarity cross-correlation. One has to be aware that the (sub)tomograms miss data due to stage limitations, which are visible as a wedge in Fourier space (missing wedge). As a result, the structure is distorted and stretched along the missing wedge. This influences the cross-correlation, which is sensitive to features caused by the missing wedge. One way to mitigate this is the application of a similar wedge-shaped binary mask in Fourier space, aligned to the missing data, on the reference. The cross-correlation will then only compare sample features (constrained cross-correlation [85]). After aligning, similar sub-tomograms can be averaged, and the alignment can be repeated with a new average. To avoid overfitting, the average should be low-pass filtered before entering the next round. The filter is based on the Fourier shell resolution of two independent datasets. If there are different structures in the sub-tomogram pool (conformers, different complex stoichiometry, etc.), the sub-tomograms should be classified. The most straightforward classification is based on the cross-correlation values from the alignment step. Cross-correlation-based classification requires a template. Classification based on principle component analysis/k means clustering (PCA) avoids this potential bias, although PCA is sensitive to the missing wedge as well. Again, the missing wedge can either be masked with a binary mask [85,86] or a 3D sample function that is based on the resolution-weighted CTF [73]. Lastly, the classification (and alignment) can be done with a maximum likelihood approach, in which CTF-corrected sub-tomograms are used [87]. Although the tomogram reconstruction, alignment and classification are described as sequential (Figure 5), many programs and workflows process some or all steps together. The power of STA has been recently demonstrated by generating <4 Å structures for in situ complexes [35] and 2–4 Å structures for ex vivo/in vitro complexes, rivalling SPA structure generation [34,35,73]. For an exhaustive discussion of the workflow and theory, see [88,89]. Programs for STA are Dynamo [86], EMAN2, emClarity [73], M [35], PEET [90], i3 [82], Pytom [91] and Relion [87].

## 4. Outlook

The advances in cryoET over the last five years have been breathtaking. Structures generated by in vitro samples have improved to the point that they are comparable in resolution to SPA structures. Although it is the exception rather than the rule, in situ structures can have a resolution that allows *de novo* model-building with atomic resolution. One of the biggest challenges in cryoET is throughput. The collection of tomograms is still limited by mechanical constraints of the microscope, resulting in dedicated tomography microscopes. Unlike in STA, data collection for SPA allows the collection of high-resolution data on 100–200 kV microscopes [92]. The production of lamella is still slow and requires an experienced user [93]. Automation of this process will make this technique more accessible to a broader user base [94,95]. The low SNR is still a problem in the data collection. Phase plates improve the contrast greatly, but the VPP comes with serious drawbacks. Recently, a report showed that a standing light wave can be used to shift the phase of the electron beam [96], which avoids the problems of the VPP.

Even if improvements in imaging instrumentation and FIB milling take place, the protein abundance (or scarcity) will limit what complexes can be imaged in situ. Instead, in situ is suited for the observation of the formation of complexes in response to stimuli. High-resolution in situ cryoET could establish a molecular etiology of diseases, like amyloid diseases [52] or mitochondria diseases [97]. The same is true for in vitro cryoET, where it could be shown that amyloid oligomers and protofilaments, but not fibrils, intercalate into one leaflet of the bilayer [98]. Lastly, microfluidic [99] and optogenetic tools [100] enable us to use cryoEM for the investigation of time-dependent processes in isolated complexes and in organisms. The latter will be accessible for cryoET with lamellae that can be lifted out of tissues [101].

## Figures and Tables

**Figure 1 ijms-22-06177-f001:**
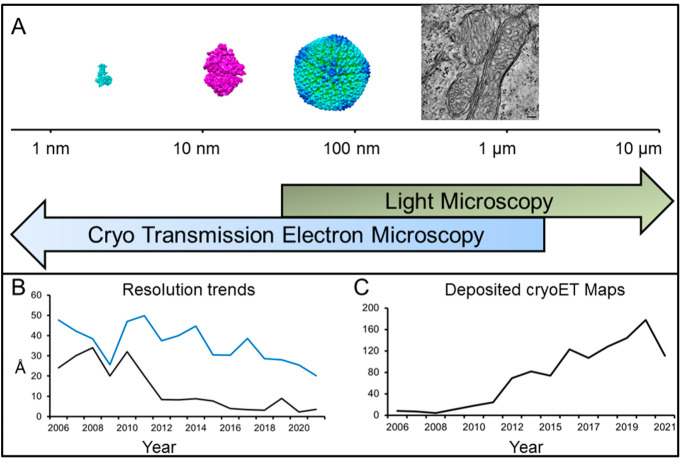
Cryo-electron microscopy is a multiscale technique with sub-nm resolution. (**A**) Electron microscopy covers a wide resolution scale. In cyan is the cryoEM structure of the α adrenergic receptor (cyan, EMDB-22357), the human ribosome is an example of a MDa molecular machine (magenta, EMDB-11796), adenoviruses (blue and green) are at the upper limit of single particle analysis (green/blue, EMDB-10768). The electron tomogram of mitochondria from the retina (EMDB-11125) is an example of an imaging regime where electron microscopy and light microscopy overlap. (**B**) Resolution range of maps solved with sub-tomogram averaging. In black is the trend for maps with the highest resolution and in blue, the average resolution. The impact of direct electron detectors around 2012 is clearly visible. (**C**) Trend of tomography maps deposited in the electron microscopy database.

**Figure 2 ijms-22-06177-f002:**
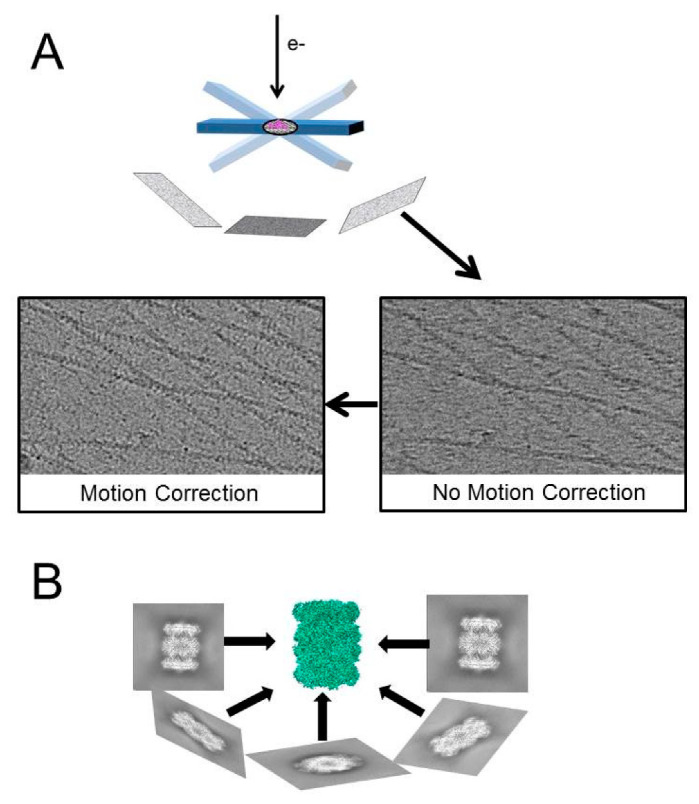
General principles in the generation of electron tomograms. (**A**) Collection of movies at different sample tilt angles. The movies are motion corrected and electron dose weighted. The left box shows the sum of the individual movie frames. Only movement correction reveals fine features of the sample (right box). (**B**) Corrected projection images of the sample with known angles yield a structure of the imaged object.

**Figure 3 ijms-22-06177-f003:**
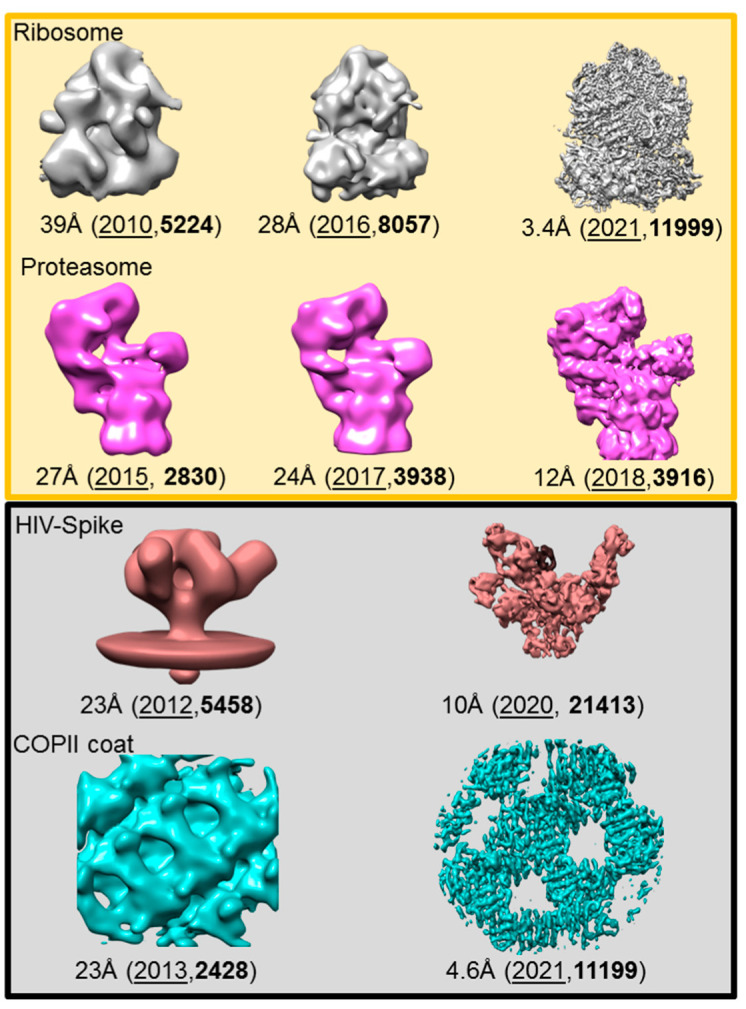
Examples of the improvement in resolution of sub-tomogram averaging maps. The resolution is given for each structure, together with the year (underlined) and the EMDB-ID (bold). The top (yellow panel) shows structures from in situ specimens. The grey box below shows samples that were reconstituted in vitro.

**Figure 4 ijms-22-06177-f004:**
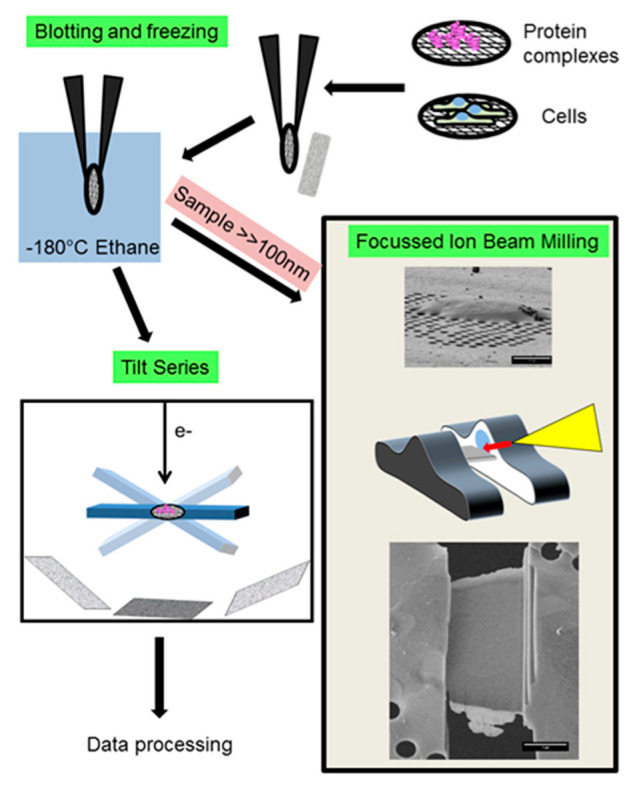
Schematic workflow for sample preparation and data collection. Samples can be isolated complexes, cell compartments or cells grown/settled on grids. After excess liquid is blotted away, the sample is plunge-frozen (high-pressure freezing is excluded here). If samples are sufficiently thin, one can proceed to collection of tilt series right away. Cells and thick samples can be processed with focused ion beam milling (beige box). The SEM image shows a cell on holey carbon. During the milling process (see middle graphic illustration), cell material is ablated by sputtering Ga2+ ions on the cell with an ion gun (yellow). As a result, we obtain a lamella thin enough for imaging (lower picture). The SEM images were kindly provided by Prof Linda Sandblad, Umeå Core Facility for Electron Microscopy.

**Figure 5 ijms-22-06177-f005:**
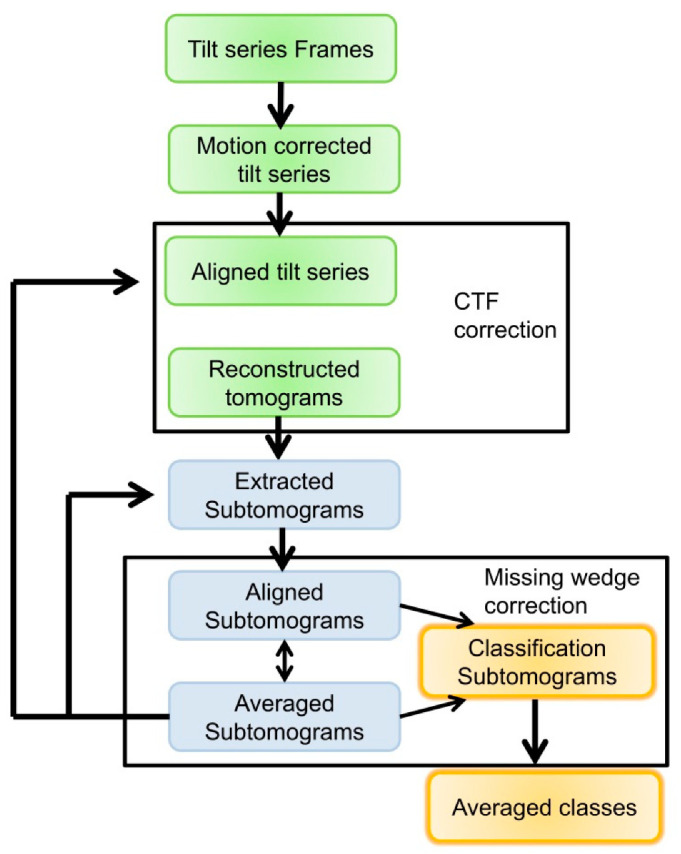
General workflow for the processing of tomograms and sub-tomogram averaging. First, the movies are motion-corrected, and the resulting images are aligned. The aligned tilt series is then used to calculate the tomogram. CTF correction can happen during these steps. The individual sub-tomograms are then identified and extracted. The sub-tomograms are then aligned against references and averaged. At this stage, missing wedge compensation is applied. Aligned sub-tomograms (or averaged sub-tomograms) are then classified. Processing is iterative (black arrows). Averaged sub-tomograms help to refine steps earlier in the workflow. It should be pointed out that this is a general workflow. Depending on the software, some steps are merged.

## Data Availability

Not applicable.

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
