# Peer review of "Coming of Age: Cryo-Electron Tomography as a Versatile Tool to Generate High-Resolution Structures at Cellular/Biological Interfaces"

_ijms, 2021, doi:10.3390/ijms22126177_

Round 1

Reviewer 1 Report

The review by Zuoneng Wang et al. is a helpful overview of current approaches in cryoET, with appropriate historical perspectives, as well as broad directions in which the field is evolving. The review should be published essentially as it is. There are a few minor typos that should be corrected.

The word “amount” should not be used for countable entities. On line 70, “amount of structures” should be “number of structures.” On lines 109-110 “a minimum amount of (tilted) views” should be “a minimum number of tilted views.”

On line 70, “For once…” should be “For one…”

Line 84 needs a comma after “For SPA related studies”

Line 332: “data collection SPA “ should be “data collection for SPA”

Author Response

Dear Reviewer,

thank you for the time and the opportunity to strengthen the manuscript. 

Please see the response below.

Reviewer #1

The word “amount” should not be used for countable entities. On line 70, “amount of structures” should be “number of structures.” On lines 109-110 “a minimum amount of (tilted) views” should be “a minimum number of tilted views.”

We have substituted ‘amount’, wherever inappropriate

On line 70, “For once…” should be “For one…”

The sentence has been changed

Line 84 needs a comma after “For SPA related studies”

The sentence has been changed

Line 332: “data collection SPA “ should be “data collection for SPA”

The sentence has been changed

Reviewer 2 Report

General comment:

This work reviews the use of ctyo electron microscopy has a tool for structural biology. The authors discuss and proposed novel approaches and pathways to follow in this research area.

The paper is well written. 

Specific comments throughout the paper:

Abstract. 

The abstract is short and respect the maximum number of words. I wonder if it can be more catchy, maybe with another sentence. This is a minor aspect.

Lines 17-18: There is no "1." section. It should be "1. Introduction". Please revise and fix. 

1.1. Structural biology of macro-molecular complexes.

Do not use finish the subsection title with ".". Please revise.

Lines 19-21: We were taught that structur and function are related (Monod, Jacques. "On chance and necessity." Studies in the Philosophy of Biology. Palgrave, London, 1974. 357-375.), however, nowadays this paradigm is questionable and its relevance is revised, please see:

Redfern, Oliver C., Benoit Dessailly, and Christine A. Orengo. "Exploring the structure and function paradigm." Current opinion in structural biology 18.3 (2008): 394-402.

Greslehner, Gregor P. "What do molecular biologists mean when they say'structure determines function'?." (2018).

I suggest the authors to consider these points in their work and accounts for them by adjusting the weights of some bald statements.

Line 69: About the "meteoric rise", as often done in review works, maybe a quantitative perspective can be provided to the readers (e.g. number of articles). Figure 1 is not recalled in this part of the text. Maybe the authors can consider to refer to Fig. 1 therein. 

Line 88: "shortcomings" - Please fix.

1.2 Electron tomography

Lines 105: Missing space between the word and the reference, please fix.

About sect. 1.2, I belive that providing a visual summary of some of these fundamental concept can severly help the readers and favor their understanding, especially for the less technical and those who will approach the topic for the first times by your work. This could add value to your work. Fig. 3 is not recalled here.

No Sect. 2, only subsections. Please revise the work.

3.2 Sub-tomogram averaging

Lines 295-296: The details about the convolutional neuronal network architectures, training and validation can be of interest. The machine learning and artificial intelligence-related works are very few. Maybe this point can be helpful for a discussion which can be of interest to a wider readership.

Lines 303-304: More details about the cross-correlation can be provided. 

4. Outlook

A table or a figure for summarizing and schematize the challengese and field of application of cryoET can be of help for the understanding of the value of your work.  

Author Response

Dear Reviewer,

thank you for your time and the opportunity to improve the manuscript. We incorporated changes and replied to the concerns voices. Please see attachment.
